# Investigating the Stability of Six Phenolic TMZ Ester Analogues, Incubated in the Presence of Porcine Liver Esterase and Monitored by HPLC

**DOI:** 10.3390/molecules27092958

**Published:** 2022-05-05

**Authors:** Leroy A. Shervington, Oliver Ingham

**Affiliations:** 1School of Pharmacy and Biomedical Sciences, University of Central Lancashire, Preston PR1 2HE, UK; 2Kindeva Drug Delivery, Bakewell Road, Loughborough LE11 5RB, UK; oliver.ingham@kindevadd.com

**Keywords:** temozolomide (TMZ), phenolic TMZ esters, hydrolysis, glioblastoma multiform (GBM), porcine liver esterase (PLE), stability, high-performance liquid chromatography (HPLC), diode-array detector (DAD)

## Abstract

Previous published data from our group showed the encouraging in vitro activities of six phenolic temozolomide (TMZ) ester analogues (ES8–ES12 and ES14) with up to a five-fold increase in potency compared to TMZ against glioblastoma multiform cell lines and TMZ-resistant O^6^-methylguanine-DNA methyl transferase (MGMT)-positive primary cells. This study investigated the stabilities of the six phenolic TMZ ester analogues in the presence of porcine liver esterase (PLE) as a hydrolytic enzyme, using high-performance liquid chromatography (HPLC), monitored by a diode-array detector (DAD). Determining the rates of hydrolysis of the esters provided a useful insight into the feasibility of progressing them to the next phase of drug development. Fifty percent of TMZ esters consisting of *para* nitro, chloro, phenyl and tolyl groups (ES9, ES10, ES12 and ES14) were hydrolysed within the first 4.2 min of PLE exposure, while the TMZ esters consisting of *para* methoxy and nitrile groups (ES8 and ES11) demonstrated increased stability, with 50% hydrolysis achieved in 7.3 and 13.7 min, respectively. In conclusion, the survival of these phenolic TMZ esters on route to the target site of a brain tumor would be a challenge, mainly due to the undesirable rapid rate of hydrolysis. These findings therefore pose a question regarding the effectiveness of these esters in an in vivo setting.

## 1. Introduction

Temozolomide (TMZ) is the standard of care treatment for patients diagnosed with glioblastoma multiforme (GBM). However, the median survival of patients diagnosed with glioblastoma is only increased by approximately 2.5 months when treated with TMZ and radiotherapy, compared with that treated with radiotherapy alone [1]. Furthermore, the current literature suggests that 60–75% of patients experience no clinical benefit from TMZ and a further 15–20% of these patients develop significant toxicity, which can make further treatment unsafe [2]. Therefore, there is an inherent need for a chemotherapeutic agent that is safer and more effective for patients with high-grade glioma [3].

A comprehensive review of the literature indicates that position 8 on the imidazole ring of TMZ holds significant promise in terms of improving the activity of the molecule [4,5,6,7,8,9,10,11,12,13,14,15,16,17]. Numerous studies have demonstrated a degree of success in investigating both ester and amide analogues of TMZ. However, to date, modifications at position 8 of the imidazole ring of TMZ have not been fully explored. The rationale behind designing analogues at position 8 is based on three concepts. Firstly, a series of ester and amide analogues of TMZ were synthesised to assess the impact of hydrogen bond-donating and hydrogen bond-accepting sites at position 8, provided by ester and amide moieties. Secondly, the inclusion of phenyl esters and amides with various aromatic substituents could provide an insight into the mesomeric and inductive effects of these analogues. Thirdly, the addition of alkyl chains to ester moieties would provide a platform to assess the effect of increasing the lipophilicity of TMZ analogues. Therefore, the initial aim of the study was to synthesise a series of novel ester and amide analogues of TMZ, with the aim of improving our understanding of the structure activity relationship at this position, thus improving the activity of the molecule [18].

Novel ester and amide analogues of TMZ were synthesised using two routes and assessed for their cytotoxic potency against the glioblastoma cell lines, i.e., U87-MG, 1321-N1 and GOS-3, and the human glial cell line, i.e., SVGp12, used as a control [18].

The antiproliferative effects of the phenolic ester and amide analogues synthesised from TMZ have been evaluated against specific glioma cell lines and patient-derived primary cultures [18]. A number of these phenolic TMZ esters analogues (ES8–ES12 and ES14) were found to have an up to five-fold increased activity against specific GBM cell lines, when compared to TMZ (Figure 1). Furthermore, the lead phenolic TMZ esters were found to induce significant antiproliferative effects against TMZ-resistant O^6^-methylguanine-DNA methyl transferase (MGMT)-positive primary cells. These findings were encouraging, since they indicated that these phenolic TMZ esters could potentially provide an alternative treatment for patients diagnosed with GBM [18].

In light of the in vitro activities of the phenolic TMZ esters analogues (ES8–ES12 and ES14), the stabilities of these analogues needed to be investigated in an environment resembling that of the physiological system. Due to the fact that these analogues are esters, a fundamental concern regarding their stability in an environment rich in esterase enzymes was posed.

The hydrolysis of these TMZ esters yield TMZ acid and the corresponding alcohols (Figure 1) and can be readily monitored by high-performance liquid chromatography (HPLC).

In mammalian systems, esterase enzymes are primarily found in the liver and gastrointestinal (GI) tract as well as in other regions of the body [19], playing a pivotal role in drug metabolism. These enzymes are responsible for the hydrolytic metabolism of numerous drug substrates, including esters, carbamates, thioesters and amides [20]. The designed ester prodrugs serve as vehicles for the active portion of the drug which offers increased lipophilicity to enhance oral absorption and cellular penetration [21]. Increased cellular uptake due to enhanced lipophilicity is thought to correlate with increased activity. Since the analogues described here were designed to act on cancerous brain tissue, it is important to ascertain the stability of these analogues when exposed to esterase enzymes and thus should provide an insight into their behavior during their potentially challenging journey towards the intentioned site of the action.

The most common carboxylesterases enzymes found in humans are human carboxylesterase 1 (hCE1) and human carboxylesterase 2 (hCE2) [20]. Mammalian carboxylesterase enzymes are found primarily in cells of the liver and GI tract but are also found in other tissues throughout the body [22]. The enzymes reside in the cytoplasm and endoplasmic reticulum, contributing to the hydrolysis of various endogenous and exogenous substrates [23]. hCE1 is mainly found in the liver, while hCE2 is found predominantly in the small intestine [24]. Although both hCE1 and hCE2 are promiscuous towards a wide range of ester substrates, the most predominate one depends on the ester substrate [20,25]. Esters containing large acyl and small alcoholic groups are the preferred substrates of hCE1, while hCE2 has a preference for esters with small acyl and bulkier alcoholic groups [22,26]. More in-depth accounts of these carboxylesterases were described by Fleming et al. [27] and Wang et al. [28].

It is worth highlighting that the use of specific individual isoenzymes may provide a less reliable representation of ester hydrolysis in vivo, and therefore, porcine liver esterase (PLE) was used since it consists of numerous isoenzymes that provide a more acceptable representation of an in vivo environment, rather than directing the study on individual hCE isoenzymes [29]. The aim of this study was to investigate the stability of selected TMZ ester analogues, i.e., ES8–ES12 and ES14, when treated with PLE. The study would provide an initial insight into their possible suitability as potential chemotherapeutic agents specifically designed for the treatment of GBM.

## 2. Results and Discussion

### Methods Validation

The standard HPLC analytical method for the analysis of TMZ traditionally centres round an acidified buffer to aid the stability of the imidazotetrazine core [30,31,32]. The composition of the mobile phases developed for TMZ analysis usually comprises of 80–95% aqueous solvents [33]. Consequently, initial attempts at developing an HPLC method for analysing phenolic TMZ esters utilised a mixture of a sodium acetate buffer and acetonitrile in a volume ratio of 80:20 at a pH of 4.5 [34]. However, since the TMZ esters exhibited increased hydrophobicity, the elution times of these TMZ esters were found to be upwards of 40 min, which was unacceptable. The composition of the mobile phase was therefore adjusted by increasing the organic phase to 40%, thus achieving a run time of 20 min.

There were no overlapping peaks between each TMZ ester and the corresponding hydrolysed products, and therefore, monitoring the rate of hydrolysis of each ester was carried out without interference (Table 1).

Each analyte described in Table 1 was tested for repeatability at a concentration of approximately 0.02 mg/mL. A summary of the coefficient of variance (CV) of 6 injections for each analyte indicated that the method achieved suitable precision for all analytes, affording a CV of less than 1.60%. The reproducibility of the analytical method was carried out by preparing six separate solutions of each analyte at a concentration of approximately 0.02 mg/mL. Each solution was run twice, and the average peak area was recorded. The CVs obtained for the six samples analysed ranged from 0.56% to 2.64%.

In order to confirm that the method gave a response that was directly proportional to the concentration of the analyte in question, linearity determinations were carried out. In Table 2, a summary of the regression equations for each analyte is included along with the coefficient of determination (R^2^). All analytes assessed exhibited R^2^ values above 0.9988, indicating an excellent relationship between the response and the analyte concentration. The graphical representations of the relationship between the response and concentration of the TMZ ester analogues and the corresponding alcohols are shown in Figure 2; Figure 3.

In order to determine the lower limit of quantification (LLOQ) and the lower limit of detection (LLOD) of the analytical method, stock solutions of each analyte were serially diluted and analysed systematically. Figure 4 provides a suitable illustration on how each determination was deduced using a combination of TMZ acid and ES8 as an example. Chromatogram A is included in Figure 4 in order to provide a comparison to highlight what was expected to observe at a high concentration of both analytes TMZ (103 µM) and ES8 (66 µM), compared to concentrations near the LLOQ and the LLOD and taking into account the apparent level of baseline noise at very low concentrations. With reference to Table 1, TMZ and ES8 had retentions times of 2.3 and 9.7 min, respectively. Chromatogram B represented the LLOQs of TMZ and ES8, determined as a result of analysing a series of dilutions. The LLOQ was confirmed by carrying out a repeated analysis (*n* = 6) of a mixture of the two analytes at the concentrations of 320 ηM and 207 ηM, respectively, achieving a CV less than 2%. Chromatogram C represented a one-in-two dilution of the LLOQ, and chromatogram D was a further one-in-two dilution, which afforded the LLOD of TMZ acid and ES8 with the concentrations of 80 ηM and 52 ηM, respectively. Although the repeated analysis (*n* = 6) at this concentration provided a clear response, the CV for each of the two analytes was greater than 2% and therefore defined as the LLOD since the responses were not consistently repeatable. Chromatogram E represented a one-in-two dilution of the LLOD concentration and clearly indicated the absence of both the TMZ and ES8 analytes. The above procedure was also carried out for determining the LLOQs and the LLODs of all analytes, and the results are shown in Table 2.

The method validation provided an appropriate proof to show that it was suitable for monitoring the hydrolysis of the lead TMZ esters using PLE. The formation of TMZ acid and the corresponding alcohol, together with the simultaneous disappearance of the TMZ ester, was a convenient approach to monitoring the hydrolytic process.

The rate of hydrolysis for each of the phenolic TMZ esters, ES8–ES12 and ES14 when treated with PLE, was determined using the described HPLC analytical method, and all the hydrolytic reactions were quenched using acetonitrile. The phenolic TMZ ester recovery was calculated by expressing the observed concentration of the ester in the control reactions as a percentage of the determined ester concentration. In order to ensure the degradation of the esters was only specific to TMZ ester hydrolysis and not the hydrolysis of the imidazotetrazine core, reactions were carried out at pH 6. The analytical method also incorporated investigating the stability of the imidazotetrazine ring whilst ensuring PLE to retain a significant level of activity, and these findings were in line with earlier work [35] that found PLE to have a broad pH optima (pH: 6–8).

An illustration of a typical enzyme reaction profile is shown in Figure 5, depicting the gradual hydrolytic degradation of ES11 and the formation of the corresponding TMZ acid and 4-Hydroxybenzonitrile. A diode-array multiwavelength detector was used in monitoring both the degradation and formation of products during the process. Monitoring the reaction at a 250 nm wavelength provided information related to the degradation of ES11 and the formation of 4-Hydroxybenzonitrile, while monitoring the reaction at 325 nm provided information on the formation of TMZ acid in addition to the degradation of ES11. Once the method was refined and verified, it was used for monitoring the degradation of the remaining five esters. 

A summary of the results shown in Table 3 indicated that an estimated 50% hydrolysis of all six TMZ esters occurred within the first 14 min of exposure. The finding that 50% of TMZ esters, namely the *para* nitro, *para* chloro, phenyl and tolyl groups (ES9, ES10, ES12 and ES14), were hydrolysed within the first 4.2 min of incubation was an unexpected outcome (Table 3; Figure 1). The TMZ ester analogues, namely the para methoxy and nitrile groups (ES8 and ES11), presented increased stability, with 50% of both the analogues being hydrolysed within 7.3 and 13.7 min, respectively. All the hydrolytic reactions were conducted at a PLE concentration of 7.85 µg/mL, compared with studies carried out by Höllerer et al. [36], who observed a 21% hydrolysis of di-(2-ethylhexyl) phthalate after 48 h of exposure using a 1.3 µg/mL concentration of PLE, while Shervington et al. reported a 24–37% hydrolysis of various chlorambucil esters over a 24 h period, after exposure to PLE at a 9.1 µg/mL concentration [37].

The evidence from these two studies suggests a possible marked fragility of the ester bonds within the phenolic TMZ esters described in this study and, as a consequence, could negatively impact on the proposed therapeutic potential of these TMZ ester analogues.

On evaluating the para-substituted aromatic substituents of the TMZ esters, one would expect the relatively strong electron-withdrawing groups, namely the nitro (ES9) and the nitrile (ES11), to experience similar rates of hydrolysis (Figure 1). However, despite ES9 and ES11 consisting of a nitro and a nitrile aromatic substituent, respectively, the rather different rates of hydrolysis of these esters were surprising, that is, 50% of the analogue ES11 was hydrolysed within a little bit shorter than 14 min, 50% of the ES9 analogue was hydrolysed in a little bit over 4 min. Although there was no apparent explanation for this discrepancy when taking into account the differences in relation to the electronic effects of these two analogues, one possible explanation could be attributed to the toxic effect of the nitrile (cyano) group towards enzymes in general [38].

Furthermore, ES10, ES12 and ES14 did not fit the expected trend in terms of the electron-withdrawing and -donating effects of the substituents. In a related study involving the ester hydrolysis of ethyl benzoates, the rate of hydrolysis was found to be directly proportional to the electronic effects of the substituents on the aromatic ring [39]. Interestingly, this effect was not observed in this investigation, and therefore, electronic differences appear to have no significant part to play with regards to the bond strengths of the ester linkage of these analogues when subjected to PLE.

## 3. Materials and Methods

Acetonitrile (HPLC grade: ≥99.9%), water (HPLC grade: ≥99.9%), acetic acid (HPLC grade: ≥99.7%), citric acid monohydrate (HPLC grade) and dimethyl sulfoxide (reagent grade) were purchased from Fisher Scientific, Horsham, UK. Sodium phosphate dibasic (HPLC grade: ≥99.0%), sodium acetate (HPLC grade: ≥99.0%), phenol (reagent grade: >99%) and *p*-cresol (reagent grade: >98%) were purchased from Sigma-Aldrich, Gillingham, UK. PLE was purchased from Sigma-Aldrich, St. Louis, MO USA. 4-Nitrophenol (reagent grade: >97%), 4-Chlorophenol (reagent grade: >98%), 4-Hydroxybenzonitrile (reagent grade: >97%) and 4-Methoxyphenol (reagent grade: >99%) were sourced from Alfa Aesar, Haverhill, MA, USA. All reagents sourced from external suppliers were used without further purification. The TMZ esters were synthesised, purified and characterised following standard procedures [18]. The purity of each synthesised TMZ ester was also estimated to be at least 95%, using ^1^H NMR.

### 3.1. Instrumentation and HPLC Method

The analytical study was carried out on an HPLC system consisting of a Jasco PI-2089 plus quaternary gradient pump, a Jasco AS-1555 intelligent autosampler and a Jasco MD-1510 diode-array multiwavelength detector. These three components were linked to a Dell Optiplex 790 computer system using a Jasco LC-Net II/ADC. Chromatograms were acquired and processed using ChromNAV 1.0 Chromatography Data System. The HPLC analysis was carried out using an isocratic reversed-phase system. The TMZ esters were separated from their esterase-mediated degradation products, namely TMZ acid and the corresponding phenolic alcohol. Samples were injected using an autosampler from crimp top sample vials using a Rheodyne injector with a 100 µL loop. Separation was achieved using a Waters Symmetry Shield RP C_18_ column (4.6 mm × 250 mm), containing particles with a size of 5 micron. The detection of TMZ acid and TMZ esters (ES8–ES12 and ES14) was carried out at 325 nm. The detection of 4-Methoxyphenol and 4-Chlorophenol was carried out at 225 nm. The detection of 4-Nitrophenol, 4-Hydroxybenzonitrile, phenol and *p*-Cresol were detected at 325, 250, 270 and 225 nm, respectively. The mobile phase consisted of a 20 mM sodium acetate buffer and acetonitrile (60:40, *v*/*v*). The pH of the resulting mixture was adjusted to 4.5 using acetic acid. The analysis of the samples was carried out over a period of 20 min at 1 mL/min at ambient temperature.

### 3.2. Preparation of the Mobile Phase and Standards Used for Validation

The mobile phase was made up in 2000 mL batches (consisting of 1200 mL sodium acetate buffer (20 mM) and 800 mL acetonitrile). The sodium acetate buffer was prepared by solubilising sodium acetate (3.282 g) in 2000 mL of HPLC-grade water to achieve a final concentration of 20 mM. To ensure imidazotetrazines to remain stable, the pH of the sodium acetate/ acetonitrile mixture was adjusted to 4.5 by the addition of glacial acetic acid. Ten milligrams of each of the sample were solubilised in the mobile phase (100 mL) to afford the concentrations displayed in Table 4. The samples were diluted to appropriate working concentrations in order to carry out the method validation. 

### 3.3. Method Validation

Repeatability, reproducibility linearity, LLOD and LLOQ were determined as part of the validation procedures, following protocols set by The International Council for Harmonization of Technical Requirements for Pharmaceuticals for Human Use (ICH) in ICH Q2 (R1) Validation of Analytical Procedures. Text and Methodology were used to validate the method [40].

### 3.4. Repeatability

Stock solutions were diluted with the mobile phase yielding approximately 0.02 mg/mL for each of the compounds listed in Table 4. Each of the prepared solutions was analysed 6 times, and the corresponding peak areas were recorded. The CVs for the 6 determinations were calculated.

### 3.5. Reproducibility 

Six separate solutions at approximately 0.02 mg/mL were prepared from each of the standards shown in Table 4, as described above. Each of six solutions was analysed twice, and the mean peak area was recorded. The CV was calculated for the 6 determinations.

### 3.6. Linearity 

The solutions of each of the standards were diluted from their respective stock solutions to afford concentrations of approximately 0.02, 0.04, 0.06, 0.08 and 0.10 mg/mL. Each sample was analysed twice, and the mean peak area was recorded. Calibration curves were obtained for each analyte (Figure 2 and Figure 3), and the coefficient of determination (R^2^) and regression equations were calculated to assess the relation between the peak area and the concentration.

### 3.7. LLOQ and LLOD

Each of the analyte stock solutions was systematically diluted, and each sample was analysed twice. This process was repeated several times, once the detector could no longer accurately quantify and detect an analyte peak. Both the LLOQ and the LLOD were determined after calculating the corresponding CV for all analytes analysed.

### 3.8. General Protocol for Monitoring the Stability of Lead Phenolic TMZ Analogues after Exposure to PLE

A disodium phosphate/citric acid buffer (pH: 6) was prepared by adding 63.15 mL of disodium phosphate (Na_2_HPO_4_·12H_2_O, 200 mM) to 36.85 mL of citric acid (100 mM). A stock solution consisting of 80 mg of each of the TMZ esters (ES8–ES12 and ES14) were prepared by dissolving in 100 mL of DMSO. Prior to preparing the esterase reactions, all reagents were incubated at 37 °C. The control reaction, in the absence of PLE, was prepared by adding 3000 µL of the buffer to the reaction vessel along with 56.25 µL of the 0.80 mg/mL stock solution. The test reaction, with enzymes, was prepared by adding 2999 µL of the buffer to the reaction vessel along with 56.25 µL of the TMZ esters. The final concentration of the TMZ esters in the reaction vessels was 14.7 µg/mL. An aliquot of PLE (1 µL), suspended in ammonium sulphate, was added to the test reaction at zero minutes. Both the control and test reactions were then incubated at 37 °C. Aliquots with a volume of 200 µL were transferred from the reaction vessels to 1 mL Eppendorf tubes at 3 min intervals for a total of 30 min. Each reaction was terminated by quenching with 200 µL of acetonitrile in an Eppendorf tube. The Eppendorf tubes were vortexed for 30 s and then centrifuged at 13,000 rpm for 10 min. The supernatant (200 µL) was transferred to HPLC vials and diluted with the mobile phase (200 µL) before analysis.

## 4. Conclusions

These phenolic TMZ esters, namely ES8–ES12 and ES14, showed encouraging in vitro antiproliferative activity when evaluated against specific glioma cell lines and patient-derived cultures [18]. In this study, however, the application of HPLC was used in order to monitor the stability of these TMZ esters and the results were very informative, since it revealed that when these esters were subjected to in vitro hydrolytic conditions in the presence of PLE, they underwent rapid hydrolysis which would equate to a serious disadvantage, bearing in mind that they were specifically designed to enhance the ability of the TME moiety to target cancerous brain tissue. From this study, it is clear that the TMZ esters would have negligible opportunity of surviving the challenging in vivo conditions without excessive hydrolytic degradation. A positive outcome of this study lies in the fact that further in vivo and in vitro studies could prove to be costly and unnecessary; however, since it has been reported that PLE has a higher hydrolytic efficacy compared human carboxylases [41], this aspect of research would certainly be a logical next step in order to further validate our findings.

## Data Availability

Not applicable.

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
