# Peer review of "Investigating the Stability of Six Phenolic TMZ Ester Analogues, Incubated in the Presence of Porcine Liver Esterase and Monitored by HPLC"

_molecules, 2022, doi:10.3390/molecules27092958_

Round 1

Reviewer 1 Report

The present manuscript entitled "Investigating the stability of six phenolic TMZ ester analogues, incubated in the presence of Porcine Liver Esterase and monitored by HPLC" by Leroy Alexander Shervington and Oliver Ingham (molecules-1670571) describes the stability test of six phenolic temozolomide (TMZ) ester compounds, which were carried out with the use of Porcine Liver Esterase (PLE) as a hydrolytic enzyme. HPLC with DAD detection was used for monitoring the stability of these compounds.

The article is interesting from an analytical and medical point of view; therefore, it should interest the reader. The present article needs to be improved in the context of scientific content, figures, and tables. The paper meets Molecules' requirements, and it can be considered for publication after the corrections stage. My current decision is a major revision. More specific comments and observations are presented below.

  1. Abstract. Please, specify what the research was about - this should be a clear summary of the study to interest the reader. The abstract should not write what was once done or what are the plans for the future. Different fonts are currently used. “ester hydrolysis of the ester” should be rewritten.
  2. The MGMT abbreviation should be clarified before use. Please check if other abbreviations are explained.
  3. When you mention that something was monitored by HPLC, you should provide information about the type of detection. Only HPLC can not be used for monitoring stability.
  4. Page 1, lines 39 and 40. This part should be expanded in the introduction to provide more detail.
  5. Page 1, lines 43, 44; and page 2, lines 45, 46. This part should appear at the end of the introduction.
  6. “:” sometimes appear in descriptions of tables and figures, and it should be removed.
  7. There is no reference to Figure 2 in the text.
  8. RSD expressed as a percentage is the coefficient of variation (CV).
  9. Figure 2. The legend on the right must be deleted. I do not understand the idea of the legend, which is under the description of the drawing; it should be connected with the figure and not be placed as a separate drawing (the same thing later in the text). Test compounds with retention times can be assigned over the peaks (then table 1 can be deleted). The description of the x-axis "Retention Time" should be replaced with "Time". There is no dependence on the retention time here.
  10. Page 4, line 119. “without interference”. On what basis do these conclusions? What can be done in the event of strong interference effects? How would you deal with them? What types of interference effects could occur?
  11. Units. Unnecessary spaces should be removed. Please convert the units from mg/100ml to mg/ml.
  12. Tables 1, 2, and 3 should be combined without the presentation of the peak area.
  13. The measurements were performed only twice; to speak of the reliability of the obtained data, it is necessary to repeat them at least three times.
  14. Tables 4 and 5 can be combined.
  15. Figures 3 and 4. Titles should be removed. Each drawing is missing a legend. Horizontal and vertical lines should be removed. The axes must be clearly marked with markers. The points should be smaller without highlighting and the lines narrower in the graphs.
  16. It is unclear how LOD and LOQ (I prefer using LOD and LOQ instead of LLOD and LLOQ) were counted? From the calibration graph? Instead of Figure 5, LOD and LOQ should be calculated from the calibration graphs. Also, Figure 5 is strangely stretched. Please consider deleting this drawing.
  17. A discussion of the shortcomings of the research conducted should be added.
  18. Figure 6. The inscriptions above the peaks are not visible. Time should be instead of Retention Time. The border is not well-positioned.
  19. "%" is written once with a value and once with a space. Please unify it.
  20. Materials and methods. Please add countries of origin. "were purchased" is used in every sentence. Please change this.
  21. Section 3.2. What concentrations were used for validation?
  22. Validation. Were single standards or a mixture used?
  23. Conclusion. Please, emphasize clearly the advantages of the research carried out.
  24. Page 12, line 312. “relationship” should be used in the context of people. Here relation will be better.
  25. References. Please adjust it according to the requirements of the journal.
  26. The last page is blank and should be deleted.

I hope that the comments presented will help improve the article.

Reviewer 2 Report

Dear autors

The manuscript is wonderful. I enjoyed reading it all praise for the idea. Do you intend to adapt in vitro test and your method for in vivo tests?

That wolud be interesting

Reviewer 3 Report

In the abstract part, the font uses is different.

At the end of line 102 I suggest to the authors to write the exactly aim of these article.

"However due to unforeseen circumstances in late March 2020 that resulted in lockdown mode of all  UK universities, due to the COVID19 pandemic, this work was put on hold". True for all the Universities all over the world but I suggest to delete it from the conclusion part. It is true that using also carboxyl esterase 1 and 2 would have brought a better overview in hydrolysis of these esters. 

I liked the the method validation used.

The porcine liver esterase used in the article was 1 microliter. Dissolved in what solvent? Water (line 327)

Round 2

Reviewer 1 Report

Dear Authors,

Thank you for your meticulous consideration of my comments. The paper has improved substantially and, to my opinion, is suitable for publication.